# Waimānalo Pono Research: Indigenizing Community-Engaged Research with a Native Hawaiian Community

**Jane J. Chung-Do** [1,2,*], **Samantha Keaulana Scott** [2], **Ilima Ho-Lastimosa** [1,3], **Kirk Deitschman** [1], **J. Kahau Vegas** [2], **LeShay Keliʻiholokai** [1], **Ikaika Rogerson** [1], **Theodore Radovich** [1,3], **Kenneth Ho, Jr.** [1], **A. Hiʻipoi Ho** [1] and **Mapuana C. K. Antonio** [2]

1   Ke Kula Nui O Waimānalo, Waimāmalo, HI 96795, USA
2   Office of Public Health Studies, University of Hawaiʻi at Mānoa, Honolulu, HI 96822, USA
3   College of Tropical Agriculture and Human Resource, University of Hawaiʻi, Honolulu, HI 96822, USA
*   Correspondence: chungjae@hawaii.edu

**Abstract:** Native Hawaiians, or Kānaka Maoli, the first people to arrive and settle on the Hawaiian Islands, developed an ecologically sustainable food system that sustained the health of up to a million people on the islands. Colonization disrupted this system as well as the healthy lifestyle and cultural practices of the Indigenous people of the Hawaiian Kingdom. Today, Native Hawaiians face pervasive health and social inequities. To build research processes that can meaningfully and sustainably address these inequities, the Waimānalo Pono Research Hui was borne from the vision and priorities of community leaders and members of Waimānalo. Using qualitative data from the annual survey conducted with Waimānalo Pono Research Hui members, the purpose of this study is to illustrate how community engagement and community-based participatory research has been operationalized within a Native Hawaiian community to yield meaningful research. Five themes emerged from the analysis related to the ʻāina (land), pilina (relationships), consent, equitable resources, and data sovereignty. These findings demonstrate the importance of imagining, creating, and implementing research processes that are shaped by community voices.

**Keywords:** Hawaiian; research ethics; indigenous methodologies; decolonizing methodologies; community-based participatory research; community engagement





## 1. Introduction

Native Hawaiians, or Kānaka Maoli, were the first people to arrive and settle on the Hawaiian Islands. As with many Pacific Peoples, it has been proven that their arrival was intentional and not an accident as White historians have tried and failed to argue. Using the knowledge they have accumulated from gathering data from their natural surroundings, including the wind, birds, stars, and the ocean, Pacific Peoples used sophisticated wayfinding skills to navigate their way throughout the thousands of islands across the Pacific. To survive on one of the most geographically isolated land masses in the world, Native Hawaiians used their skills as data scientists and continued to collect data about the ʻāina (land) of Hawaiʻi to meticulously build sophisticated food systems, which fed and sustained up to a million people on the islands for centuries.

### 1.1. Colonization of the Hawaiian Islands

With western imperialism spreading across the Pacific starting in the early 1800s, Native Hawaiians faced forced assimilation by western missionaries and settlers and eventual illegal US occupation of the Hawaiian Kingdom in 1893 (Kameaʻeleihiwa 1992; Trask 1984). Traditional practices, including hula, surfing, lāʻau lapaʻau (traditional medicine), and ʻōlelo Hawaiʻi (the Hawaiian language) were suppressed and banned. Western imperialism also changed the concept of ʻāina. Native Hawaiians viewed the ʻāina as a spiritual kin

that provided for them through a reciprocal familial relationship. This conflicted with the western perspective that land is a commodity that can be privately owned and exploited for capital gain. American businesses took advantage of Hawai'i's tropical climate to grow sugar, pineapples, and other produce and began building mass agricultural plantations for extraction and export. These plantations diverted huge amounts of natural flowing waters from streams and local farmers and families practicing subsistence living (Goodyear-Ka'opua et al. 2014). These plantations also depleted nutrients in soil and displaced Native Hawaiians from their homes. This dispossession from land and traditional culture was further exacerbated with the US militarization of the Hawaiian islands in the 1900s as the Hawaiian Islands were seen as a strategic location in the Asia-Pacific region. In addition, western settlers brought infectious diseases to the Hawaiian islands, which Native Hawaiians had no immunity against. These diseases decimated the population, leading to a 90% population decline within a century (Blaisdell 2001).

*1.2. The Impacts of Historical Trauma and Contemporary Injustices on Native Hawaiians*

Today, Native Hawaiians continue to face systemic and societal challenges, which inhibit their optimal health and well-being. Mass agricultural plantations have been replaced by extractive tourism and real estate industries as well as the continued presence of the US military. Today, Hawai'i is often portrayed as an idyllic paradise to visitors with the tourism industry garnering $17.8 billion each year as the largest economic sector of the islands (University of Hawai'i Sea Grant Program 2018). Proponents of the tourism industry argue that tourism provides over 200,000 jobs for Hawai'i's residents. However, tourism in Hawai'i has been long criticized as reinforcing colonial structures by contributing to the degradation of the Hawaiian culture as well as the environment and natural resources (Darowski et al. 2007; Matsuoka and Kelly 1988; Trask 2000). Furthermore, the tourism industry has been accused of commodifying, appropriating, and prostituting the Hawaiian culture by selling hula, lu'au, surfing lessons to attract visitors, which implies that Hawaiian culture is good enough to sell, while Native Hawaiians face multiple barriers to practice and sustain their own culture. Hawai'i is home to some of the most diverse ecosystems in the world, but with over 10,000,000 tourists visiting Hawai'i each year, there have been numerous negative consequences for the wildlife on both land and in the ocean due to pollution related to overdevelopment and overpopulation. In addition, the US military continues to occupy approximately 150,000 acres of the Hawaiian Islands, including controlling roughly 80,000 acres on the most populated island of O'ahu, which is a staggering 22% of the island's limited land area (Kajihiro 2000). Military presence has brought substantial consequences to Native Hawaiians, particularly in the desecration of 'āina. Despite ongoing community protests, the US military has decimated 'āina across the Hawaiian Islands for bombing practice and combat training, which has further severed the relationship between Native Hawaiians and their land (Blackford 2004; Niheu et al. 2007). In addition, Native Hawaiians must compete for home ownership with non-Native military members who have increased access to the rental and housing market through VA loans and housing allowances (Pape 2015).

With continued land dispossession and structural inequities, Native Hawaiians continue to have very little access to their traditional foods and subsistence lifestyle, resulting in disproportionate risks of food insecurity and chronic diseases. For example, Native Hawaiians as well as other Pacific Islanders die at higher rates of coronary heart disease, stroke, congestive heart failure, cancer, and diabetes than the average resident of Hawai'i (Balabis et al. 2007; Look et al. 2013). Native Hawaiians are less likely to live in neighborhoods with healthy food grocers (Lee et al. 2012; Mau et al. 2008) and have less economic means of buying healthy food, with Hawai'i being one of the most expensive places to live in the US (Office of Planning, Department of Business, Economic Development and Tourism 2012). With a total of 14.8% of Native Hawaiians and Pacific Islanders living in poverty compared to 9% of non-Hispanic white residents while the cost of living continues to exponentially increase, it is becoming impossible for many Native Hawaiians to live in

their ancestral homes (Office of Minority Health Resource Center n.d.). Compared to the State as a whole, Native Hawaiians have a harder time paying for essentials, such as housing, utility bills, medicine, child care, and food (Palakiko et al. 2021). Given the exorbitant cost of living, Hawai'i has the highest rate per capita of homelessness with more than 50% of all the houseless individuals identifying as Native Hawaiian or Pacific Islander (U.S. Department of Housing and Urban Development 2017). Given these structural barriers, it is not surprising that Native Hawaiians are more likely to deal with mental health issues, such as depression, anxiety, substance use, and suicidal behaviors, which are also linked to poor nutrition and food insecurity (Hishinuma et al. 2018; Look et al. 2013).

### 1.3. The Revitalization of Native Hawaiian Practices and the Community of Waimānalo

Despite these forces of oppression, Native Hawaiians have survived and thrived. Since the 1970s, the Native Hawaiian Renaissance Movement has made tremendous efforts and strides in revitalizing Native Hawaiian cultural practices, such as the Hawaiian language, oceanic voyaging, food cultivation, and land and ocean restoration (Goodyear-Ka'opua et al. 2014). Numerous Native Hawaiian immersion schools have been established, thereby dramatically increasing the number of Native Hawaiian speakers. Communities across the islands have come together to restore land devastated by invasive species, overpopulation, and overdevelopment (Crabbe 2007). Many rural communities that are predominantly Native Hawaiian have been instrumental in retaining and promoting Native Hawaiian cultural practices.

Waimānalo, is an example of a rural predominantly Native Hawaiian community that upholds the cultural practices of Native Hawaiians. Waimānalo is located on the eastern side of the island of O'ahu and is home to approximately 7000 residents with one-third being Native Hawaiian. More than 600 Native Hawaiian families live on the Waimānalo Hawaiian Homestead. Hawaiian Homesteads were created by The Hawaiian Homes Commission Act of 1920, a federal law that was introduced into the United States Congress by Prince Jonah Kūhiō Kalaniana'ole with the intention to return land to Native Hawaiians and revitalize traditional practices of self-sufficiency. (Andrade 2022). However, to live on Hawaiian Homestead, homesteaders must demonstrate that their blood quantum is 50% or more Hawaiian, which stems from a racist system that the federal government placed onto Native American tribes to limit their citizenship (Trask 1991).

Prior to colonization, Waimānalo was a thriving community with an abundance of natural and cultural resources that sustained human and natural life. Waimānalo, which translates to "sweet water," was known for its lush waterways that fed and sustained the plants, animals, and people in the community and the wind called Limulipu'upu'u that comes from the ocean that smells of limu (seaweed). Sociopolitical events driven by western forces militarized and colonized Waimānalo. For example, the Waimānalo Sugar Plantation diverted natural waterways away from families and local farmers to increase sugar production and profits for western plantation owners. When the plantation industry began dwindling in Hawai'i, plantation owners leased the land to the US military (Bellows Air Force Station Hawai'i n.d.), which contributes to the ongoing tensions between the Waimānalo community and the US military today. Waimānalo has many strengths including its natural resources, a history of community advocacy, and a large number of Native Hawaiian grassroots organizations (Keli'iholokai et al. 2020).

### 1.4. Research in Native Hawaiian Communities

Although research has been used as a tool to address these health and social inequities, conventional western methodologies have not meaningfully engaged Indigenous communities in the research process, including Native Hawaiians. The long history of helicopter research, defined as "any investigation within the community in which a researcher collects data, leaves to disseminate it, and never again has contact with the tribe," (Cherokee Nation n.d.) by researchers who are outsiders of the "target" community, has led to exploitation and ethical breaches, thereby perpetuating colonialism and power imbalances

(Fong et al. 2003; Goering et al. 2008). Most of these "target" communities have been Indigenous communities and communities of color who face racism, oppression, and social injustices and have vastly different lived realities from the privileges and power that most researchers hold through their educational status and academic institutions (Burnette and Sanders 2014). As with other Indigenous communities and communities of color, there is a long history of unethical and exploitive research conducted on Native Hawaiians. For example, in 1866 to 1969, Native Hawaiians with Hansen's disease were forcibly exiled to a remote northern peninsula on the island of Moloka'i where unethical medical studies were conducted by government physicians to identify the mode of disease transmission without consent or regard for this vulnerable population (Au 2015). Today, community members continue to express their frustration with researchers "taking from the community without giving back to it," being disrespectful of cultural protocols, holding damaging stereotypical notions of their community, and providing no perceivable benefits to the community (Kaopua et al. 2017, p. 30). To shift the power imbalances and colonial paradigm of research, many Native scholars, community leaders, and their allies are advocating for decolonizing and indigenous methodologies (Look et al. 2013; Smith 2012; Townsend et al. 2016). These approaches have the potential to transform the deeply entrenched distrust that communities hold toward researchers, institutions, and the research enterprise to create equitable partnerships that lead to real, tangible, meaningful, and sustainable benefits to the community.

*1.5. The Waimānalo Pono Research Hui*

The Waimānalo Pono Research Hui is a community-academic partnership formed in 2017 by community leaders of Waimānalo and their longtime academic partners (Chung-Do et al. 2019). The initial support stemmed from the Detroit Urban Research Center (Coombe et al. 2017; Coombe et al. 2018). Pono means 'just,' 'balance,' and 'righteous.' Hui means 'group.' Building on the previous collective grassroots work of key community leaders, Waimānalo community members and researchers who have been working with the community for years were invited to a series of informal monthly gatherings to share a meal and "talk story." Prior to 2017, many research projects had taken place in Waimānalo with varying degrees of community involvement. The main goals of these gatherings were to (a) identify community preferences and priorities to inform future research and programming initiatives, and (b) build and strengthen relationships within the Waimānalo community, and between the community and academic researchers. Initially, there was no other agenda beyond these two goals. The direction and long-term goals were kept open to allow for participatory and organic processes to unfold. Upon meeting for several months, a clear mission with a set of priorities emerged from the iterative discussions. Under the guidance of the kūpuna (elders) of the community, the mission of this group was collectively agreed upon as the following: "to collaborate and work toward a healthier Waimānalo through education, aloha 'āina (love of the land), and honoring and transferring 'ike (knowledge) and values of our kūpuna (elders) to our keiki (children) through pono research principles" (Chung-Do et al. 2019, p. 5).

To ensure that research can truly benefit the community, identifying and centering community priorities was critical. Through multiple iterative discussions during the gathering, the top three priorities that emerged and identified included: lā'au lapa'au (traditional medicine), 'ai pono (healthy eating), and limu (seaweed) restoration. Community members expressed their desire to learn more about lā'au lapa'au, which was once banned by western healthcare institutions. The kūpuna shared stories of how lā'au lapa'au has healed them and their family, and community members shared that they have been attending various lā'au lapa'au workshops that have been initiated throughout the island. However, they yearned to learn more and emphasized the importance of placed-based practices of lā'au lapa'au. The community also recognized that the Native Hawaiian community faces multiple chronic diseases that impede their wellness and wanted more programming that emphasize healthy eating through a Hawaiian lens that use and promote traditional food

sources. Related to medicine and food, the kūpuna also raised concerns about environmental degradation and pointed out how practicing lāʻau lapaʻau and ʻai pono requires healthy land and ocean. They observed that limu was diminishing on the shores of Waimānalo and expressed their desire to restore limu as it is an important source of food and medicine for Hawaiians.

The purpose of this article is to use qualitative data that have been gathered from the members of the Waimānalo Pono Research Hui through an annual partnership evaluation survey to illustrate how community engagement and community-based participatory research (CBPR) has been operationalized within this Native Hawaiian community to yield meaningful research. Waimānalo Pono Research Hui is provided as an example of how communities can be engaged to shape the research that happens in their communities.

## 2. Methods

The membership of the Waimānalo Pono Research Hui is composed of community members from keiki to kūpuna to emphasize multigenerational learning. It is open to any Waimānalo resident and attendance is fluid. Monthly gatherings have been held since February 2017 where research proposals are discussed and updates on community events, programs, and projects are provided. Dinner is always served and the meeting ends with a hana noʻeau (traditional arts and crafts) led by one of the members of the Waimānalo Pono Research Hui who receives an honorarium for sharing their expertise. The monthly gathering attracts 15–40 people and often include families as well as representatives from the Waimānalo Hawaiian Civic Club, The Kūpuna Association, The Waimānalo Homestead Association, and Nā Wāhine Council. Members also include academic researchers and students primarily from the University of Hawaiʻi at Mānoa who have built long-term relationships with the Waimānalo community. Every year, all members are asked to complete a survey that includes quantitative and qualitative questions for continous evaluation. The latest survey was completed by 43 people with ages ranging from 17–73 years old. A total of 81% were female and 19% were male. The majority were Waimānalo residents (77%) and Native Hawaiian (77%).

The survey conducted in 2022 was administered as an online anonymous survey that was composed of 19 closed and open-ended questions. Questions asked for basic demographic information including age, sex, residence, as well as their role in the Waimānalo Pono Research Hui. Open-ended questions asked members to define what Pono Research means to them, if they feel that their voices are heard, and what their thoughts are about the Hui and the gatherings. The link to the survey was emailed to the Waimānalo Pono Research Hui email listserv in August 2022. Two reminders were given during the gatherings and by email. The survey was expected to take 15 min to complete based on past surveys. The research team, comprised of both community and academic members of the Waimānalo Pono Research Hui, extracted and analyzed the qualitative answers from the survey by identifying common emerging codes and themes related to the conceptualization of research from the Waimānalo Pono Research Hui members's perspectives. Final themes were validated through group consensus and quotes that illustrate the theme were selected and are highlighted below.

## 3. Results

*Theme 1: Indigenizing Research through the ʻāina*

Waimānalo Pono Research Hui builds on participatory approaches, such as CBPR, which aims to build authentic relationships and establish trust between academics and communities by promoting equitable partnerships (Israel et al. 2005). We also intentionally embrace Indigenous approaches to actively and meaningfully engage communities in decolonizing the research process, such as the Indigenous Research Agenda and the ʻĀina Aloha research framework. The Indigenous Research Agenda re-centers the goal of the research process in self-determination of Indigenous peoples (Smith 2012). The ʻĀina Aloha research framework, which is specifically centered in Native Hawaiian values, emphasizes

the interdependent relationships between people and the ʻāina (land) through the values of mālama ʻāina (to care for, protect and maintain land), puʻuhonua (a safe place, a sanctuary), and laulima (many hands working together towards a specific goal) (Maunakea 2015). Thus, the Waimānalo Pono Research Hui focuses on research that is action-oriented, collective, protective of ʻāina and intellectual rights, accountable, and relevant. One member stated, "*It gives opportunity for us to work on projects or programs that will enhance Ohana* [family], *as well as, helping our community to be self- sufficient, and the care of our Aina and ocean.*" Waimānalo Pono Research Hui also holds community work days where we come together to cultivate the ʻāina and nurture our relationships with one another and the land. Waimānalo Pono Research Hui was also where the Waimānalo Limu Hui (WLH) was borne, which focuses on the health of our ocean. Waimānalo Limu Hui is a community-driven initiative to bring back limu to the shores of Waimānalo. The kūpuna recalled abundant limu found on Waimānalos beaches, which have greatly decreased today because of environmental degradation and climate change. Because limu plays an important role in Native Hawaiian culture in terms of diet and lāʻau lapaʻau, WLH holds regular limu planting days on Waimānalo's shores. Importantly for this community, indigenizing research required a social justice approach that prioritized the community and applied a critical lens to identify potential harm, not just to individuals but to communities and the ʻāina. Indigenizing research meant increasing community capacity by building on community strengths, knowledge, and perspectives and focusing on producing benefits and findings that are meaningful and of importance to the community (Minkler 2010; Wallerstein and Duran 2010; Braun et al. 2012).

*Theme 2: Pilina as the Heart of All Successful and Effective Research Partnerships*

To put these values into practice, the Waimānalo Pono Research Hui created Protocols and Rules of Engagement to promote transparency and accountability (Keaulana et al. 2019). These protocols highlight the importance of pilina or relationships, community consent and culturally centered approaches, equitable resources and benefits, and data sovereignty and dissemination. The Waimānalo Pono Research Hui members collectively understand that at the heart of all successful and effective research partnerships is pilina, which can be loosely translated as "relationships, connections, fitting, adhering to one another." Any research projects proposed to the community may be declined because it is not safe or relevant. Before researchers or anyone pitches an idea, they must SHOW UP and take the time to get to know the community and build pilina. This allows everyone to get to know each other beyond their labels and instead, as complex human beings. When asked what they enjoy the most about the Waimānalo Pono Research Hui, one member stated, "*It allows me to stay connected to the community and to learn of all the good things that our programs do for Waimanalo. It also allows me to see everyone.*" To highlight the importance of relationship and trust-building, the Waimānalo Pono Research Hui requires that all participants join at least three meetings or community activities before proposing an idea to the Hui. In addition, when an idea is brought forth to the community, only those who live in Waimānalo are allowed to vote on the proposal, which keeps power within the community.

*Theme 3: Consent Must Be Interwoven in Every Step of the Research Process*

Consent, both at the individual and community levels, is expected to be interwoven in every step of the research process, from conceptualization of the research idea to data dissemination. The Waimānalo Pono Research strives to ensure all members are aware of their rights and know what questions to ask about any research including our own. When asked if they know their rights when asked to participate in research, all of the respondents stated that they do. For example, one member stated, "*Yes. I know I have rights to say yes to things I believe in and no to things I'm not sure of or don't think will make a real community impact.*" Another member stated, "*[I can} state how I do/don't feel comfortable with the data being used; my expectations about communicating results back to me, etc.*" Thus, we always ensure that the default is to always ASK, and never assume. The process of consent includes discussing and gathering input from the community, even if it is harsh or negative. Researchers must have the humility and openness to take criticisms, learn from them, and do better. In addition, it is the responsibility of researchers to ensure the community fully understands

what they are consenting to. This process is necessary to ensure that the approach is truly centered in the values, practices, priorities, and strengths of the community.

*Theme 4: Recognize the Value of Community Knowledge with Equitable Resources*

Equitable resources means that the community is properly compensated for their time, knowledge, and role in the project. As one participant stated, this will help, "*make sure research and programs and any initiatives are always wanted and embraced by the community and will have lasting benefits.*" It cannot be assumed that the community are willing to be unpaid volunteers while researchers and their students get paid. If possible, community members should have the opportunity to have paid roles in the project and be an active part of the research team, such as a co-Principal Investigator. Ideally, if a project is to be truly community-driven, the Principal Investigator should be from the community and the funds should run through the community if capacity exists for this. Researchers can lend their expertise to the project but the ultimate decision-making power is within the community. To build this capacity in Waimānalo, a nonprofit organization, called Ke Kula Nui O Waimānalo, was started by seven members of the Waimānalo Pono Research Hui in 2017. With community permission, Ke Kula Nui O Waimānalo now houses and owns the research and programs that have been developed by the community, for the community (Ho-Lastimosa et al. 2020). For example, the Waimānalo Limu Hui is a program under Ke Kula Nui O Waimānalo. All data collected for this program is maintained and managed by Ke Kula Nui O Waimānalo. Anyone who would like to access the data must ask for permission from Ke Kula Nui O Waimānalo with the final approval from the Waimānalo Pono Research Hui. Another component that was identified by the community is the cultural importance of food in gathering in Indigenous communities. This has been an issue in the past because some funders do not allow the purchase of food. Recognizing the importance of sharing a meal in Hawaiian culture, every proposal that is pitched to the Waimānalo Pono Research Hui must include food and it is the responsibility of the person pitching the idea to make this possible if the funder does not allow it.

*Theme 5: Data Sovereignty Is Vital in Telling Our Own Stories*

Data sovereignty is a critical aspect of decolonizing research and data and promoting pono research, which was defined by a member as "*ethical research that is developed with, by, for Kānaka in Waimānalo.*" The Waimānalo Pono Research Hui Protocols and Rules of Engagement state that the data resulting from the project must be owned by the Waimānalo Pono Research Hui. That means other researchers who wish to access the data and conduct a secondary data analysis must first ask the Waimānalo Pono Research Hui for permission and also cultivate a relationship with the community. Even though the data may be de-identified, every study must be approved by the Waimānalo Pono Research Hui, due to the risk of group harm. In addition, the community should also be the first to know the results of the study and the data should be shared back with them first and foremost. The community should be continuously updated at every step, especially when the data is being interpreted to ensure the interpretations are accurately grounded in the community's lived experiences. In the data dissemination stage, the community should be lead or co-authors so they can tell their own story. For far too long, the stories of disenfranchised communities have been told by outsiders, who are often white, male, and with more institutional privileges. This has not only led to errors in the stories that are being told, but have also led to sacred stories being shared without permission, and stories that perpetuate racist and sexist stereotypes that often portray these communities as "lesser than." Therefore, it is important to the Waimānalo Pono Research Hui that we tell our own stories.

## 4. Discussion

For far too long, Native Hawaiians have endured the long-standing impacts of colonization and historical trauma, yet they continue to resist the adversities that stem from these socio-cultural determinants of health, demonstrating their ability to thrive and survive in their ancestral homelands. This study is another demonstration of the importance of strengths-based, community-driven, and culturally grounded approaches to research in

the context of Hawai'i and with Indigenous communities. In particular, the Waimānalo Pono Research Hui was conceptualized as a community-academic partnership that aims to identify community priorities of research, inform future research and programs, and foster relationships between the Waimānalo community and academic researchers at large. This study reports on the qualitative themes from an annual survey distributed to the Waimānalo Pono Research Hui listserv and based on results from the August 2022 survey. Five major themes resulted including the importance of Indigenizing research through the 'āina, acknowledging pilina as the heart of all successful and effective research partnerships, interweaving consent in every step of the research process, recognizing the value of community knowledge with equitable resources, and promoting data sovereignty and telling our own stories.

These findings advance the growing literature on indigenous methodologies and the call for transforming the conventional paradigm of research ethics that is truly grounded in community voices. Our research demonstrates the importance of collectively creating structures that engage communities at the grassroots level. It is not enough to merely include community, the Waimānalo Pono Research Hui is driven and led by the community. This enhances sustainability of initiatives while most projects tend to end once the funding ends. For example, after the initial seed funding from the Detroit Urban Research Center was spent, the community members, especially the kūpuna, insisted that we continue to meet regardless of funding. This indicates the strong level of community buy-in and the importance of the space created by the Waimānalo Pono Research Hui. In addition to the formation of the Waimānalo Limu Hui that was described earlier in this paper, the discussions and priorities identified by the Waimānalo Pono Research Hui has led to several other initiatives. For example, with 'ai pono (healthy food and healthy eating) being identified as one of the priorities, we have garnered funding for two 'ai pono programs that promote food sovereignty, MALAMA Aquaponics and Ulu Pono Mahi'āina. MALAMA Aquaponics is a program and a research study that teaches multigenerational Native Hawaiian families how to build and use a backyard aquaponics system to grow their own food to make healthy dishes and traditional medicine from the herbs, vegetables, and fish grown in the system (Ho-Lastimosa et al. 2019). Families complete nine hands-on skill-building workshops over six months and build and take home an aquaponics system at the end of the program. Multiple positive impacts of health have been documented (Beebe et al. 2020). Ulu Pono Mahi'āina teaches families to apply agroforestry principles to create food forests and gardens in their backyards. A total of 115 community members from 27 families completed the program and installed 8 Polynesian food forests in their yards, which is expected to produce 50,000 lbs of food within five years. Both program emphasize growing traditional crops (e.g., kalo, māmaki, 'olena, nī'oi, uala) that are often difficult to find in mainstream grocery stores or are exorbitantly expensive. In addition to teaching families how to grow their own food in a variety of ways, these programs also integrate lā'au lapa'au, which is the third priority identified by the Waimānalo Pono Research Hui. Families are taught how to make traditional Hawaiian medicine using the herbs and plants that they grow.

Numerous studies have demonstrated the importance of 'āina in the wellness with indigenous people (Antonio et al. 2020; McGregor et al. 2003). Therefore, it is imperative to understand how each community define wellness and identify the important components that shape their health. Because the health and connection to 'āina has always been at the forefront of the conversations of the Waimānalo Pono Research Hui, many of our initiatives integrate this wholistic way of thinking to restore the health of Native Hawaiian communities. Land is seen as living breathing entity by many indigenous peoples with its own rights (Māori Law Review 2014). When thinking of consent and rights of people, we must also consider the impacts on the health of the land. The importance of relationships and trust-building has been highlighted by multiple studies (Chung-Do et al. 2016; Thaman 2008; Christopher et al. 2011). Processes and protocols developed by the Waimānalo Pono Research Hui ensure the identification of health and research priorities through a commu-

nity lens, which redistributes the power and privilege of research and ensures that the community remains in the driver's seat of research agendas (Chavez et al. 2007). This also supports the importance of taking a community-driven and culturally grounded approach to health and research, which has been shown to effectively reduce health inequities that exist for communities with a history of oppression, colonization, and historical trauma (Burnette and Sanders 2014).

**Author Contributions:** Conceptualization, all; methodology, J.J.C.-D. and S.K.S.; validation, all; formal analysis, J.J.C.-D. and S.K.S.; investigation, resources, data curation, J.J.C.-D., S.K.S., I.H.-L., K.D., J.K.V., L.K. and I.R.; writing—original draft preparation, J.J.C.-D., S.K.S., M.C.K.A., T.R. and K.H.J.; writing—review and editing, visualization, J.J.C.-D., M.C.K.A. and A.H.H.; supervision, J.J.C.-D., I.H.-L., K.D., all; project administration, all; funding acquisition, J.J.C.-D., I.H.-L. and K.D. All authors have read and agreed to the published version of the manuscript.

**Funding:** This research received no external funding.

**Institutional Review Board Statement:** The study was conducted in accordance with the Declaration of Helsinki, and approved by the Institutional Review Board of the University of Hawaiʻi at Mānoa (protocol code 2019-00508 and 28 June 2019) for studies involving humans.

**Informed Consent Statement:** Informed consent was obtained from all subjects involved in the study.

**Data Availability Statement:** The data presented in this study are available on request from the corresponding author. The data are not publicly available to follow the Research and Protocol Guldelines of the Waimānalo Pono Research Hui.

**Acknowledgments:** Mahalo nui loa to the members of the Waimānalo Pono Research Hui, Detroit Community-Academic Urban Research Center, Kamehameha Schools, Castle Foundation, and Papa Ola Lōkahi for their support.

**Conflicts of Interest:** The authors declare no conflict of interest.

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
