# Peer review of "Waimānalo Pono Research: Indigenizing Community-Engaged Research with a Native Hawaiian Community"

_genealogy, doi:10.3390/genealogy6040090_

Round 1
Reviewer 1 Report
Kia ora
Many thanks for the invitation to review the paper. It was an excellent read, providing a good foundation/context for the research and I think it will be useful for other Indigenous communities who wish to lead/partner/participate in research.
General comments:
1. Citations start as numbering then another referencing style is used – please be consistent.
2. The discussion was very short. It provided a summary of the themes and demonstrated links with already published work but I would have liked to see:
a. how these findings are unique and add to our knowledge/understanding
b. evidence for successful outcomes as a result of research Waimānalo Pono research, especially with regards to the three priorities (traditional medicine, health eating and limu restoration).
A few specific suggestions here:
Page 1
· Line 7 remove were
· Line 8 remove who
· Line 9 remove ‘Forces of’
· Line 10 capitalise Indiegnous
· Line 12, borne not born
· Line 15 use full wording for CBPR
· Line 26 capitalise White
· Line 39 gap between Hawaiian and language
· Line 43 change men to people or businesses
Page 2
· Line 83 remove second ‘traditional’
· Line 86/7 add something linking these conditions to poor nutrition
· Line 117 full stop after first ‘Native Hawaiians’. Then “More than 600…live on….”
· Line 123 reference?
· Line 124 remove organizing
· Line 132 remove subsistence
Page 4
· Line 161 reference Smith Decolonising Methodologies
· Lines 170, 71 and 74 change quotations marks to inverted commas
· Lines 183-6 need reference if quoting
Page 5
· Line 201 gap between and and ocean
· Line 206 use full wording for CBPR when introducing it for the first time
· Line 211 remove ‘everyday’
· Line 214-5 consider change to “where research proposals are discussed and updates on …. Are provided”.
· Line 219 spelling of Civic
· Line 22 change to to with
· Line 229 change included to including
· Line 230 change ask to asked
· Line 232 seems to be a large space
· Line 233 unsure what litserv means. Also says August 2022 here but 2020 on line 227 – please check
· Line 240 change to Results
· Line 243, if introduced earlier don’t need full wording for CBPR here
· Line 249 capitalise Indigenous
Page 6
· Line 260 change born to borne or consider using started or commenced
· Lines 265-9 change to “Importantly for this community, indigenizing research required a social justice approach, prioritized community, applied a critical lens to the potential harm of research to both individuals and to the wider collective, and would contribute to developing community capacity, building on community strengths, knowledge and perspectives”.
· Line 276 culturally?
· Line 281 change to “any research proposal to the community may be declined because it is not safe or relevant”.
· Lines 281-91 used the word ‘pitch’ a lot here. Consider other wording eg brought, considered, proposed
Page 7
· Line 320 add organization after non-profit?
· Lines 321-2 “houses and owns” – please explain/elaborate
· Line 324 – capital for Indigenous

Reviewer 2 Report
Theme 1: Indigenizing research through the ʻāina Theme 2: Pilina as the heart of all successful and effective research partnerships Theme 3: Consent must be interwoven in every step of the research process. Theme 4: Recognize the value of community knowledge with equitable resources Theme 5: Data sovereignty is vital in telling our own stories
Clear, succinct and visionary understanding of the potential of Indigenous thinking in the rejuvenation of research within communities and society. I appreciate the lucid summary of Hawaiian history that dovetailed within a contemporary response. This was an honor to read. The key for this kind of research was indeed linking people to the healing of land/ ocean/water. This form of mutual causality activates pilina/healing/growth in our lives. Brilliant and necessary. I look forward to sharing this article with our classes and future scholar-practitioners.
